# Brief communication: Weak control of snow avalanche deposit volumes by avalanche path morphology

Hippolyte Kern[1], Nicolas Eckert[2], Vincent Jomelli[1,3], Delphine Grancher[1], Michael Deschatres[3], Gilles Arnaud-Fassetta[4]

[1]Université Paris 1 Panthéon-Sorbonne, LGP-CNRS, 1 place Aristide Briand, Meudon, France

[2]UR ETNA, INRAe, Université Grenoble Alpes, Grenoble, France

[3] Aix-Marseille University, CNRS, IRD, Coll. France, INRAE, CEREGE, 13545 Aix-en-Provence, France

[4]Université de Paris, UMR 8586 PRODIG, 8, rue Albert Einstein, 75013 Paris, France

*Correspondence to*: Hippolyte Kern (hippolyte.kern@lgp.cnrs.fr)

Abstract: Snow avalanches are a major component of the mountain cryosphere, but little is known about the factors controlling the variability of their deposit volumes. This study investigates the influence of avalanche path morphology on c. 1500 deposit volumes recorded between 2003 and 2018 in 77 snow avalanche paths of the French Alps. Different statistical techniques show a slight but significant link between deposit volumes and path mean elevation and orientation, with contrasted patterns between winter and spring seasons. The limited and partially non-linear nature of this control may result from the combined influence on the genesis of deposit volumes of mean path activity, climate conditions and mechanical thresholds determining avalanche release.

## 1. Introduction

Snow avalanches are a major component of the mountain cryosphere (Beniston et al., 2018) that often put people, settlements and infrastructures at risk. These gravitational processes result in rapid and complex snow flows (Mc Clung and Gauer, 2018). Despite significant progresses over recent years regarding the mechanical behavior of snow in motion and related flow regimes (Köhler et al., 2018), critical aspects of snow avalanche dynamics remain less known, such as the factors controlling deposit volumes. The latter limited knowledge is surprising as snow avalanche deposit characteristics determine the damage and disturbance to people, buildings and communication networks (Leone et al., 2015). Previous work documented the sedimentological characteristics of snow avalanche deposits (Jomelli and Bertran, 2001; Bartelt et al., 2009). It has also been observed that snow characteristics may vary from the starting zone to the deposit area (Jomelli, 1999; Jomelli and Bertran, 2001). Research conducted on experimental sites in Switzerland (Sovilla et al., 2015; Kölher et al., 2018) or from Canadian, Japan and European Alps field survey (Mc Clung and Gauer, 2018) showed weak links between avalanche deposit size, path slope and avalanche maximum frontal speed. However, morphological factors driving the volumetric characteristics of avalanche deposits have never been explored in a systematic way on a basis of a large data set. Here, the objective is to exclusively examine the relationship between avalanche path morphology and snow avalanche deposit volumes. Using an exceptional sample of 1491 snow avalanches deposits documented from 2003 to 2018 on 77 avalanche paths from the French Alps and using simple (descriptive) to advanced (deep learning) statistical techniques, we present a first detailed quantification of how avalanche path morphology impact snow avalanche deposit volumes. Specifically, we show that the control of deposit volumes by path morphology is weak but significant.

**2 Data and methodology**

**2.1 Avalanche deposit volumes**

This research was conducted on the upper part of the Maurienne valley in the Northern French Alps between the municipalities of Lanslevillard and Bonneval-sur-Arc (Fig. 1). Because of its important number of active avalanche paths, this study area is particularly relevant for our analysis (e.g., Eckert et al., 2009; Favier et al., 2014; Zgheib et al., 2020). The dataset of snow avalanche deposit volumes used in this study is primarily based on the Permanent Avalanche Survey (EPA) which was created at the beginning of the 20th century to document avalanche events as exhaustively as possible on more than 3,000 avalanche paths in the French Alps. For each single avalanche event, the geometric size of the deposit is documented, based on a visual estimate carried out by devoted survey operators from the EPA network. For each deposit, the length, the width and the mean depth is estimated, which eventually provides a volume estimate. The EPA operators are very familiar with the studied paths, including their snowpack-free morphology and systematically use the same predefined observation point, so as to maximize the accuracy of the estimation, especially the depth of the deposit. The depth of the deposit remains however difficult to estimate as for safety reasons this is not based on direct measurements on the deposit. This is especially problematic in case of pre-existing successive deposits, but observers try to take such effects into account as much as possible when providing their visual estimates. For each path, EPA operators systematically use the same predefined observation point, so as to maximize the accuracy of the estimation, each deposit is estimated individually in order to avoid carrying out an estimation on several stacked deposits. However, a further correction and completion work was carried out to develop a more comprehensive and error-free snow avalanche deposit database (Kern et al., 2020). Input errors or outliers introduced within the EPA when old records registered on paper archives were converted to numerical data were spotted and corrected. In addition, few other snow avalanche events (less than 1% of the total number of events) that were not included in the EPA were added from other sources: operational services in charge of hazard management and a citizen science dataset (data-avalanche.org).

From the entire deposit volume data set available since the beginning of the 20th century (Kern et al., 2020), our study only uses data covering the period from November 2003 to June 2018 (15 "full" avalanche seasons). This limits the biases and inaccuracies induced by the estimation method which were much higher earlier due to less sharp topographical references available to the EPA operators and a less standardized observation protocol until the 2000s (Kern et al., 2020). Thus, our study includes 1491 single avalanche events and associated deposit volumes registered in 77 distinct paths (Fig. 1). A small part of the avalanches are preventively triggered to protect the road network. According to the EPA database only 53 of the 1491 avalanches were triggered by explosive. Also, few defense structures are present in the studied paths but not enough to significantly affect our analysis. All in all, avalanche activity in the study area is among the most natural ones still existing in the French Alps.

To analyse the possible links between path morphology and deposit volumes, we first computed the interannual mean deposit volume in each of these paths. Then, the same operation was done for both the winter (avalanches that occurred between 1st of November and 29th of February) and spring (avalanches that occurred from 1st of March to 31th of May) sub-seasons. Eventually, in order to investigate the potential influence of avalanche activity on deposit volumes, we evaluated the interannual mean number of avalanches per year and path, including within the computation snow avalanche records for which we did not calculate volumes. Seasonal (Nov-Feb and Mar-Jun) avalanche occurrence rates were also evaluated.

The data from two weather stations handled by Météo-France and located at elevations of 1715 m a.s.l. and 2740 m a.s.l. in Bessans for the period 2003-2017, respectively (Fig. 1), was analyzed in order to determine climate conditions having locally prevailed over the study period. This showed that the depth of the local snowpack regularly exceeds 50 cm at 1715

78    m a.s.l. and 200 cm at 2740 m a.s.l.. The winter (Nov-Feb.) season is characterized by a cold mean air temperature (-4°C

79    at 1715 m a.s.l., -5.5°C at 2740 m a.s.l.), with heavy precipitation that nearly only fall in the form of snow but the mean

80    depth of the snowpack remains relatively thin (90 cm at 2740 m a.s.l.). By contrast, the spring season is characterized by

81    higher mean air temperatures (3.5°C at 1715 m a.s.l., -2°C at 2740 m a.s.l.) and the occurrence of significant daily warm

82    spells (daily mean air temperature up to 25°C at 1715 m a.s.l), which favors the occurrence of rain on snow events and

83    wet snow avalanches. The mean daily fresh snowfall is half as much as during the winter season, but, the mean snowpack

84    remains thick (170 cm).

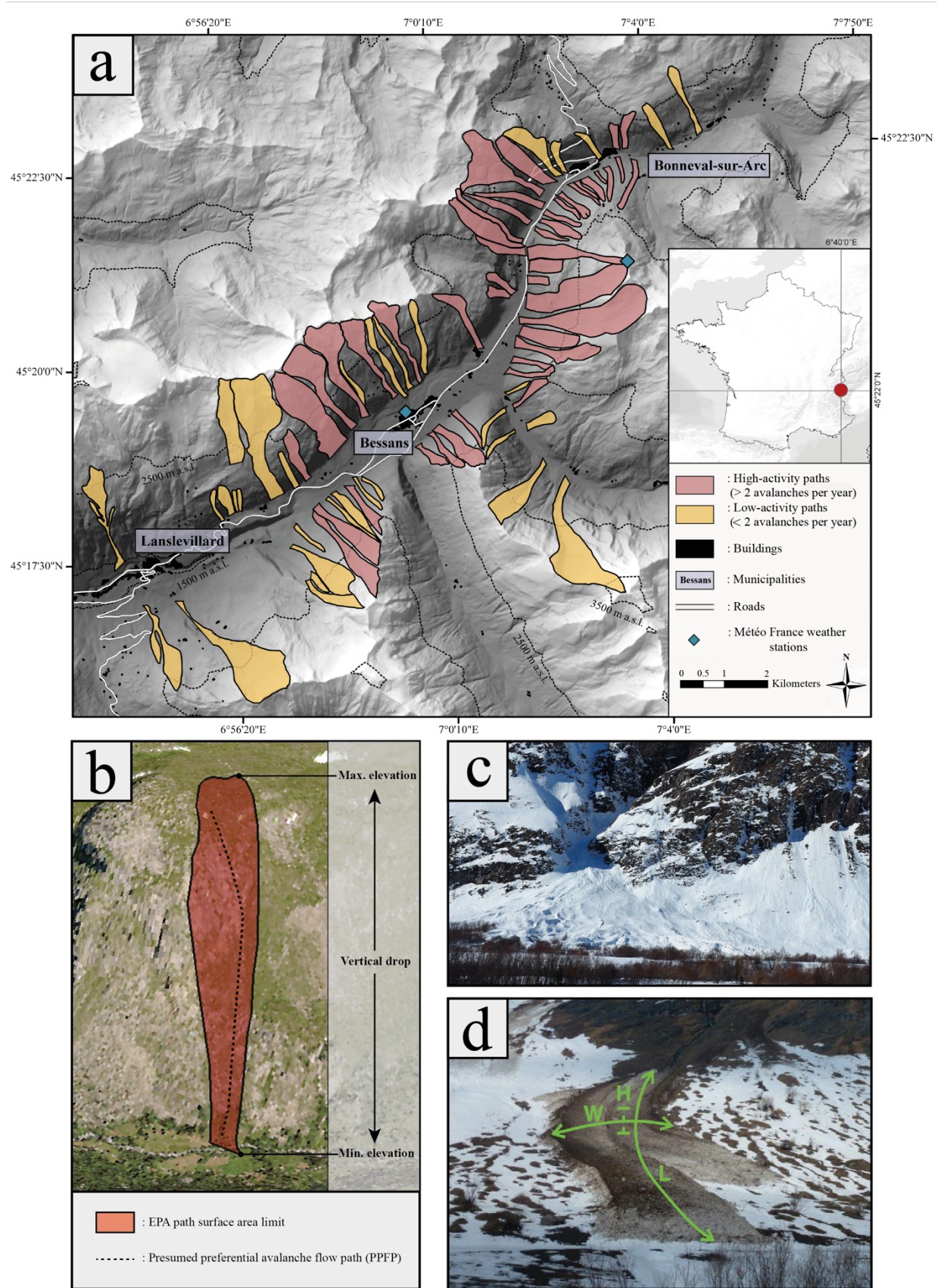

**Figure 1: (a) Study area: snow avalanche paths from the EPA database and avalanche activity according to our completed database in the upper part of the Maurienne valley, French Alps, between 2003/04 and 2017/18; (b) Example of morphological characteristics of an avalanche path from the EPA database (path n°44, Bessans); (c) Snow avalanche deposit in Bessans (© INRAE ETNA, 2018); (d) Method for visually estimating the deposit volume, H: height W: width L: length (© INRAE ETNA, Bessans, 2019)**

## 2.2 Avalanche paths morphology and related volume samples

For each studied EPA path (Fig. 1), a large set of morphological variables was calculated from a 1 meter accuracy DEM. We first defined the presumed preferential avalanche flow path (PPFP) within the path. The PPFP is the simplified thalweg of each path. For each PPFP, the length, the min, max and average slope were calculated as well as the min, max and average elevation and the vertical drop. From the whole extent of each EPA path, surface area, min, max and average slope were calculated (Supplementary Table 1). In addition, the primary orientation of each path was evaluated as a categorical variable with 8 possible values corresponding to the 8 cardinal directions: N, N-E, E, S-E, S, S-W, W and N-W. For quantitative analyses, this categorical variable was further transformed into a vector of 8 binary variables. Namely, a path got assigned 1 for the binary variable corresponding to its primary orientation (e.g., North), and 0 for the 7 other binary variables corresponding to the 7 other directions.

The studied paths are characterized by a mean elevation of 2281 m a.s.l varying from 1936 m to 2942 m a.s.l. Concerning the dimensions the paths, the mean vertical drop is 950 m a.s.l and the surface area varies from 3 to 172 ha. The mean path slopes vary from 26° to 49° with a mean slope of 39°(Supplementary table 1). The paths are mostly oriented either South or North-West. None of the paths present a North-East orientation. (Supplementary figure 1).

## 2.3 Statistical analyses

One-way analyses of variance (ANOVA) were first conducted to evaluate the significance of the partition into two subsamples (winter & spring deposits, and "high frequency paths" with more than 2 events per year versus "low frequency paths" with less than 2 events per year). In other words, we investigated whether or not the variability of deposit volumes by path morphology varies i.) according to the season, and hence, prevailing snow and weather conditions and related types of avalanche activity, ii.) according to path's mean activity.

To shed light on the control of deposit volumes by avalanche path morphology, Spearman correlation coefficients (r) were first calculated between each descriptive variable of path morphology and the annual (Nov-Jun.), winter (Nov-Feb.) and spring (Mar-Jun.) deposit volume data series. This coefficient was chosen rather than the more classical Pearson one because the statistical distributions of deposit volumes are asymmetric, with extreme values strongly departing from the mean (Fig. 2). With a dataset of 77 individuals (one mean deposit volume per avalanche path), the relationship is significant at the 0.05 level if the Spearman coefficient is greater than 0.25 in absolute value.

Stepwise linear regressions were undertaken in order to determine the combination of morphological variables that best explain the variability of mean deposit volumes. This was done first using the complete database of 77 paths and 1491 deposit volumes. Afterwards, distinct linear models were fitted i) on the 649 snow avalanche deposits recorded in 68 of these paths during the Nov-Feb. winter season, and ii) on the 842 snow avalanche deposits recorded in 73 of these paths during the Mar-Jun. spring season. With a stepwise procedure, the set of predictive variables retained is selected by an automatic sequence of Fisher F tests. Starting from an initial null model with no covariates and then comparing the explanatory power of incrementally larger and smaller models, it combines forward selection and backward elimination. Forward selection tests the variables one by one and includes them if they are statistically significant based on the p-value of the F statistics, while backward elimination starts with all candidate variables and tests them one by one for statistical significance, deleting any of them that are not significant on the basis of the p-value of the F statistics. We used the classical 0.05 and 0.01 probability thresholds for forward selection and backward elimination, respectively. However, before running the stepwise regression, a variable preselection was completed. This was made to avoid too much redundancy within potential predictors, which can lead to masks and numerical instabilities during the stepwise selection. To this aim, Pearson's correlation ρ was calculated between all pairs of potential morphological variables (Supplementary

Table 2). Among the strongly correlated variables ($\rho > 0.8$ and $p < 0.001$), we kept as potential predictor only the one
with the highest marginal correlation with deposit volumes.
Eventually, in order to account for potential nonlinear relationships between morphologic variables and snow avalanche
deposits, more flexible neural networks models were constructed, again both for the full data set and the winter/spring
sub-seasons. For the three data sets, the full set of 16 morphological variables previously presented was used as potential
covariates (8 quantitative variables and the 8 binary variables corresponding to orientations). Both standard 3-layers and
advanced 8-layers (deep learning) neural networks were developed. Models were trained using 70% of the data randomly
selected from the analyzed sample of paths/mean deposit volumes with the Levenberg-Marquard algorithm (Moré, 1978).
Validation was carried out with 15% of the data and model testing was carried out with the remaining 15% of the data.
This typical machine learning approach allows both progressive improvement of the model with cross validations and
limitation of overfitting. To account for the variability of obtained relations, a 100 bootstrap iterations was conducted,
varying data partition into calibration/validation/test subsamples and initial conditions for the Levenberg-Marquard
algorithm.
**3 Results**
**3.1 High spatiotemporal variability of deposit volumes and avalanche activity**
High spatial variability in deposit volumes is observed, with the path mean deposit volume over the study period varying
from 1400 to 49,800 $m^3$, the "mean of the mean" path deposit volumes being 15,100 $m^3$ (Fig. 2). If one looks further in
the distribution of mean deposit volumes, the sample mean and dispersion is significantly higher (one-way ANOVA $p =$
$0.010$) for winter season (path mean deposit volume $= 18100 m^3$) than for spring season (path mean deposit volume $=$
$12847 m^3$).
Concerning the temporal variability, both the years 2003 and 2004 recorded particularly small mean deposit volumes ($<$
$4000 m^3$). On the other hand, 2006 and 2014 recorded particularly large mean deposit volumes, with annual means of
$35,800 m^3$ and $47900 m^3$, respectively. Moreover, a substantial proportion of the largest deposit volumes occurred in

154 2017.

A strong variability in avalanche activity is observed between 2003 and 2017, with 30 low active paths ($<2$ avalanche
events per year) and 47 active paths ($>2$ events per year). On average, about 96 events with documented deposit volumes
are triggered per year in the study area. The avalanche year 2017 was particularly active, with 526 events with documented
deposits. Some of the paths are particularly active and show more than 35 events over the studied period. Paths located
at Bonneval-sur-Arc and Bessans show more avalanches than those at Lanslevillard, in the lowest elevation part of the
study area. Avalanche activity is more abundant in spring season (860 avalanches with documented deposits) than in
winter season (631 avalanches with documented deposits). Considering the frequency indicates that the high frequency
paths show larger mean deposit volumes ($16800 m^3$) than low frequency paths ($12900 m^3$). This observation is validated
by a one-way ANOVA ($p = 0.029$). A significant relationship exists between winter deposit volumes and the mean annual
frequency of each path ($r = 0.35$; $p < 0.001$).
**3.2 Relationships between path morphology and deposit volumes**
Avalanche deposit volumes are significantly correlated with several morphological variables and with a South-East
orientation. For the full (annual) data set, the best pairwise correlations are with path mean elevation ($r = 0.51$), surface
area ($r = 0.48$) and max elevation ($r = 0.46$). However, a clear distinction between the two seasons is observed (Table 1).
For the winter season, the correlations are significant ($r > 0.25$) between deposit volumes and seven morphological

variables among which mean elevation and surface area are the best predictors. The winter deposits reveal a significant correlation with an East orientation. In addition, deposit volumes are also influenced by frequency, through the negative correlation of frequency with min slope ($r = -0.24$ $p < 0.05$) and positive correlation with max slope variables ($r = 0.51$; $p < 0.05$). By contrast, correlations between path morphological variables and deposit volumes are significant in spring season for 5 variables only, and correlations are lower. Slope variables are not significantly correlated with deposit

| | Min elevation | Max elevation | Vertical drop | Mean elevation | Min slope | Max slope | Mean slope | Surface area | N-W | N | N-E | E | S-E | S | S-W | W |
|---|---|---|---|---|---|---|---|---|---|---|---|---|---|---|---|---|
| Annual | **0.31** | **0.46** | **0.41** | **0.51** | **-0.31** | **0.24** | -0.05 | **0.48** | -0.20 | 0.11 | | 0.15 | **0.28** | 0.16 | 0.05 | 0.02 |
| Winter | **0.38** | **0.52** | **0.48** | **0.55** | **-0.40** | **0.25** | -0.12 | **0.52** | -0.21 | -0.08 | | **0.28** | 0.22 | -0.17 | -0.02 | 0.19 |
| Spring | **0.28** | **0.35** | **0.27** | **0.43** | -0.24 | 0.19 | 0.10 | **0.34** | -0.01 | 0.07 | | -0.05 | 0.19 | 0.16 | 0.06 | -0.09 |

volumes for the spring season.

**Table 1: Spearman correlation r between morphologic variables and avalanche deposit volumes. Values in bold are significant at the 0.05 level.**

Stepwise linear regressions (Fig. 2) highlight the combined effects of morphological variables on deposit volumes. For the three analyzed data sets (annual and winter/spring) none of the variables related to the PPFP is selected because of low or non-significant correlation values. By contrast, all selected variables are relative to the path surfaces: min elevation, mean elevation, min slope, max slope, mean slope, surface area and orientation. Max elevation and vertical drop were removed as they were too strongly correlated. For the annual data set, the retained model includes three positive significant morphological variables increasing the deposit volumes: mean elevation and North and South-East orientation. However, $R^2$ remains low with only 30% of the deposit volume variability explained by these variables. The seasonal stepwise linear regression shows interesting differences between the two seasons. The retained models include four significant morphological variables increasing the deposit volumes for winter season: mean elevation and east, south-east and west orientation. Only one positive variable is retained in the model for spring season: mean elevation. Resulting $R^2$ is higher for deposits in winter ($R^2=0.41$) than for spring deposits ($R^2 = 0.15$), which remains particularly low (Fig. 2).

Neural networks significantly enhance the predictive power with higher $R^2$ values between the full set of morphological variables and deposit volumes, for both annual and seasonal data sets. With the 3-layer models, depending on the bootstrap iteration, best 2.5% of the models reach $R^2$ of 0.46 (Supplementary Table 3), and, again, best fit is obtained for winter season ($R^2=0.57$ for the 2.5% best models, versus 0.37 for the spring season). Switching to the even more flexible deep-learning based 8-layer models even enhances these values to $R^2=0.76$ and 0.54 for the 2.5% best models in the winter and spring season, respectively. However, on average on the 100 bootstrap iterations, retained neural network models do not reach high predictive power. For instance, the median $R^2$ value among the 8-layer models fitted on the annual sample is 0.28 only. By contrast, as soon as a reasonably good agreement between observations and prediction is obtained (Figure 2 on which models providing $R^2=0.61$, $R^2=0.58$ and $R^2=0.34$ are showed), discrepancies are low all over the calibration, validation and test samples, with a nearly unbiased, Gaussian-like, distribution of residuals (Supplementary figure 2). This all confirms the weak but significant control of deposit volumes by morphological variables. The increment in predictive power with regards to linear models also suggests that this control relies at least partially on non-linear relationships.

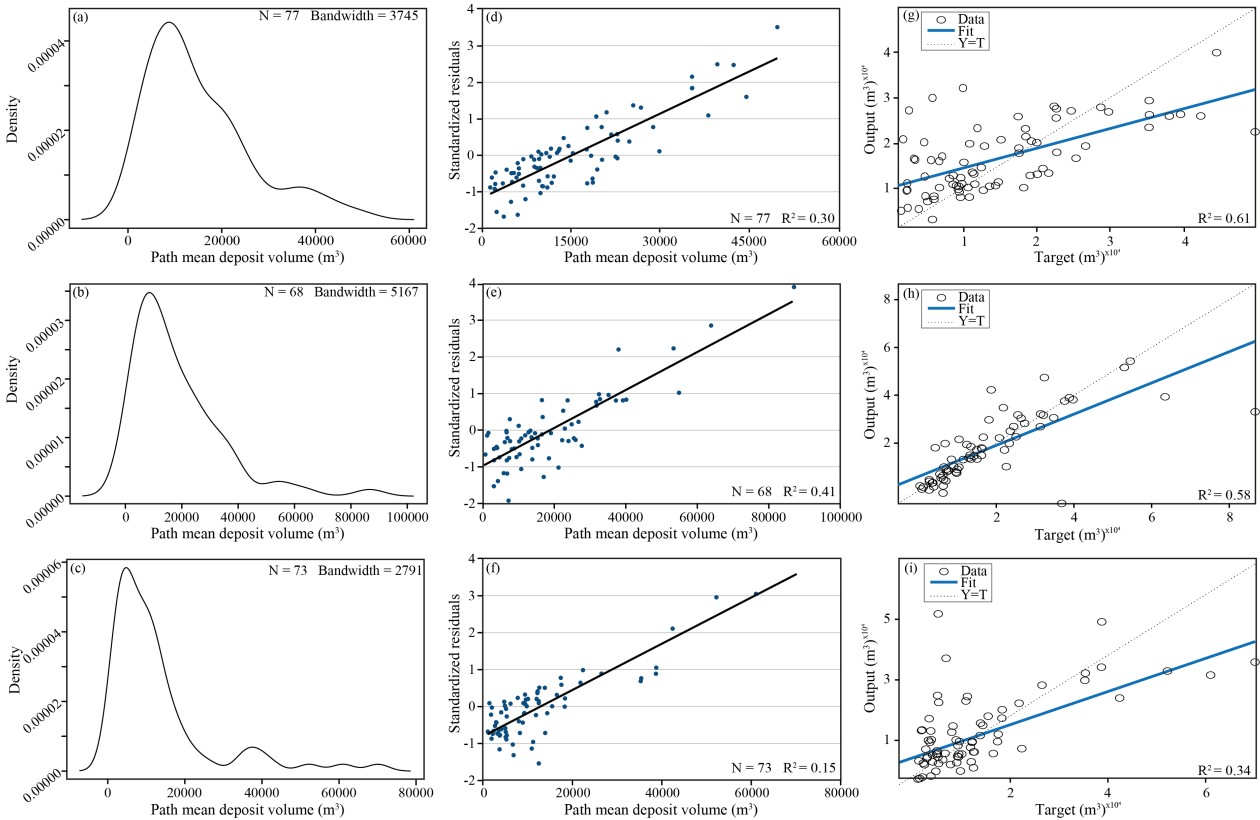

Figure 2: Statistical characteristics of snow avalanche deposit volumes. Kernel density estimation of mean path deposit volumes at the annual Nov-Jun. time scale (a), in winter, Nov-Feb. (b) and in spring, Mar-Jun. (c); Standardized residuals of stepwise linear regression results between path mean deposit volumes and path morphological variables for annual (d), winter, (e) and spring (f), linear correlation between observed deposit volume and values predicted by one neural network for the annual (g), winter (h) and spring (i) data set (Y= predictor and T=target).

## 4 Discussion, conclusion and outlooks

In this study, using a unique dataset from 77 paths located in the upper part of the Maurienne valley, we explored the influence of snow avalanche path morphology on deposit volumes. Using descriptive statistics, we showed a significant positive relationship between avalanche path morphology and the mean deposit volume at the path scale. The best simple relationships were observed with path mean elevation (r = 0.51) and surface area (r = 0.48): a large surface at high altitude favors important snow accumulation and large deposits. The seasonal subsampling analysis revealed differences in the strength of the correlation between volumes and paths morphometric variables, with higher values for winter than for spring. This may be due to climate conditions that strongly control spring deposit volumes (e.g. wet snow avalanches are released as soon as cohesion drops within the snowpack due to the apparition sufficient liquid water, and rather independently of the snow mass. Only winter deposits show a weak correlation with an orientation: East (r = 0.28). This correlation shows that winter deposit volumes may be influenced by prevailing climatic conditions. Specifically, we suspect that the significant influence of orientation reveals wind impacts. Thus a prevailing wind from the west during the winter season may cause large accumulations of snow on the east oriented hillside, later favoring important deposit volumes. Such hypothesis remains however speculative without direct wind measurements at high elevations.

Linear regression did not improve the relationships much, with no more than 30% of the annual deposit volumes
variability explained by a combination of morphological variables, increasing to 41% for winter deposit volumes
variability but decreasing to 15% of the spring deposit volumes variability. In the three cases, mean elevation is retained
as a relevant predictor, which underlines the relevance of snow availability in relation to elevation concerning the
determination of deposit volumes. Orientation variables are only retained by the annual and winter deposits model. Winter
deposits show a strong positive relationship with East, Southeast and West path orientations. This indicates how important
the solar radiation and/or the path positioning in respect to the prevailing wind direction may be to generate the snowpack
and then produce instabilities, later influencing volume deposits. However, there is no reliable data on wind direction or
speed at the scale of a massif, so it is not possible to precisely characterize the wind contribution to our study. The use of
a more flexible neural network approach leads to significant improvements, notably with some deep learning-based
models, but, overall, the power of morphological variables to predict snow avalanche deposit volumes remains somewhat
limited. In light of these results, we suggest that path's morphology controls deposit volumes significantly but weakly,
and at least partially on the basis of non-linear relationships. This could be confirmed (or not) with further studies in
different mountain areas where topography and/or avalanche activity regime is different. Additional morphological
descriptors, such as convexity or concavity of the starting zone, could slightly improve the predictive power of the models.
However, we suspect that no matter which descriptors are used, the control of the deposits volume by path morphology
remains weak.
Mean avalanche frequency appears as an important factor to explain these results. Indeed, slope variables partly influence
the annual frequency and indirectly the deposit volumes. High frequency paths (> 2 events per year) present a steeper
slope than low frequency paths (< 2 events per year) paths: 40° and 37°, respectively. Also, high frequency paths show
larger path's mean deposit volumes (16,800m$^3$) than low frequency paths (12,900m$^3$). These somewhat counterintuitive
results are in line with those of Sovilla et al. (2010) that highlighted a negative correlation between slope angle and deposit
depths, partly affected by the avalanche activity.
We interpret the weak relationship between mean path deposit volumes and morphological variables to be partly due the
predominant control of avalanche activity by snow mechanical behavior. This especially occurs trough the mechanical
thresholds involved in avalanche triggering processes (Gaume et al., 2012, Li et al., 2020), which are primarily related to
snow depth and stratigraphy in the release area as well as to the slope and ground roughness in the release area. This may
explain why snow avalanche deposit volumes do not seem that much affected by avalanche path size, for example. Also,
mechanical release thresholds may explain the significant variations we observed in the control of winter or spring
deposits by path morphology, since, from one season to another, different snow depths and stratigraphy may lead to
release for different slopes / elevations as soon as the critical stress value is exceeded. Differences in the snowpack
characteristics may also explain why the winter deposit mean volumes present more important values. Indeed the winter
snowpack is less stable and prone to large avalanche triggering, in other words snow storms are frequent in winter and
favor major instabilities and large snow avalanches. Note that we did not take into account in our study the roughness of
the ground, which was not possible to accurately document over the full sample of paths, but this could be an insightful
perspective for further work.
More widely, we speculate that the weak relationship between volume and morphological variables may be due to an
important control by climate conditions since variations in snowpack characteristics determine avalanche triggering and
flow properties (Steinkogler et al., 2014, Kölher et al., 2018), and notably snow entertainment and deposition during the
flow, which ultimately determines deposit volumes. Recent climate change thus impacts snow avalanches frequency,
magnitude, seasonality and localization, leading, e.g., to an increasing proportion of wet snow avalanches documented in
the French Alps between 1958 and 2009 (Naaim et al., 2016). Our approach should therefore now be extended to
simultaneously take into account the control of deposit volumes by morphological and meteorological variables on a
wider study area, and how these controls evolve as climate change goes on. Such an approach combining morphological
and climatic variables has, for example, already been applied in Svalbard (Eckerstorfer and Christiansen., 2011) or for
debris-flows in the French Alps (Jomelli et al., 2019).

## Data Availability

The whole EPA avalanche data is freely availed at https://www.avalanches.fr/. The dataset of mean deposit volumes and
morphological variables analyzed in this study can be requested to HK.

## Author contribution:

VJ and NE designed this research. MD provided the EPA dataset. HK, VJ, NE and DG performed the analyses. HK wrote
the manuscript on the basis of the input of all co-authors.

## Competing Interests

The authors declare no conflict of interest.

## Acknowledgements

Hippolyte KERN holds a PhD grant from Université Paris-1 Panthéon-Sorbonne. This work has received financial support
from LabEx DynamiTe (ANR-11-LABX-0046) as part of the "Investissement d'Avenir" program and from INRAE,
member of Labex OSUG@2020. The numerous people from ONF-RTM and INRAE that contributed to the EPA survey
with the financial support of the French Ministry of the Environment are acknowledged.

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
