# Peer review of "Brief communication: Weak control of snow avalanche deposit volumes by avalanche path morphology"

_The Cryosphere, 2021_

## Referee Comment (RC1)

Review of "Brief communication: Weak control of snow avalanche deposit volumes by paths morphological characteristics"

General Comments

The authors present a study that examines the influence of select morphological variables on avalanche deposit volumes in 77 avalanche paths located in the French Alps from 2003 to 2018. They apply a variety of statistical techniques to examine which variables have the most influence and compare the efficacy of each technique to determine the one with the best predictive power.

The manuscript is organized and presented well but could use a final grammar/wording revision to improve the readability in places. The sequence of statistical analysis from the ANOVA to the stepwise regression techniques appear appropriate for this dataset and sample size. The interpretation is mostly clear (see specific comments/questions below). I understand that the purpose of this study is to examine morphological variables exclusively. I think to emphasize this the authors should explicitly state this as the objective in the Introduction. My biggest concern is the limited scope of inference that using just morphological variables to predict deposit volumes provides. However, I think the authors clearly state this limitation in the Abstract and Discussion and this work provides a solid quantitative measurement of the influence of morphological variables on deposit volume. It seems this work falls into the Brief Communication manuscript type by reporting on "novel aspects of experimental methods and techniques which are relevant for scientific investigations within the journal scope". Therefore, I recommend publication after the specific comments below are addressed.

Specific Comments
Line 14: The last sentence is a bit confusing in the way it is currently written. What do you mean by "weakness"?

Line 28: Include "morphological" before "factors" to emphasize the use of morphological variables exclusively.

Line 29-30: Should read "snow avalanche deposits". Remove "volumes".

Line 41: Geometric size or destructive size?

Line 42: How was the depth of debris deposits calculated, specifically? Width and length seem relatively straightforward to estimate, particularly based on images after the event, but can you elaborate on how observers estimate depth from the designated vantage point(s)?

Line 59: Do you mean "including snow avalanche records for which we did not calculate volumes"?

Line 60: Avalanche occurrence rates?

Line 63-69: The reporting of snow depth in this paragraph is a bit confusing to me. Is the mean annual snowpack at the end of February only 90 cm at 2740m? Then, on average, the snowpack

height increases another 80 cm from March through the end of May to reach 170 cm? Line 63 states that the snowpack depth at this elevation regularly exceeds 200 cm, though. Please clarify.

Line 79: What is the exact accuracy of the DEM? 1m?

Line 82: It seems that the shape of the starting zone (or thalweg of the starting zone) represented by convexity and/or concavity might be a useful variable as well. Did you examine this?

Line 116: Change "carryout out" to "completed".

Line 130/Table 1: This table should probably be moved to the Results section.

Line 141: Is this statistically significant? If not, consider using 'substantial' to avoid confusion.

Line 147: stronger? Perhaps 'more abundant' is a better word choice.

Line 150-151: I assume that each avalanche deposit volume is treated/measured individually as opposed to measuring the cumulative debris volume in paths with >1 avalanche over the course of the winter. Please clarify either here or in the Methods section how the individual avalanche height component in each volume calculation was separated from the cumulative height in paths where subsequent avalanche debris "stacked" on top of older debris.

Line 153: See my previous comment on Table 1 and since you reference the values in Table 1 here, Table 1 should be moved to the Results section.

Line 158-159: text reads "...positive correlation of frequency with min slope (r = -0.24 p < 0.05). This is slightly confusing because the r value is negative indicating a negative correlation. I read this negative r value as such: as frequency decreases, minimum slope increases. It seems that what you mean is that higher frequencies are correlated with lower minimum slope angle. Is this correct? Please clarify for the reader.
Also, perhaps I missed it, but I don't see frequency related to the morphological variables in any table in the main manuscript or supplement. Please clarify and consider including frequency correlation values in the table as well.

Line 161: change "slopes" to "slope"

Line 201/Figure 2: Please define 'Y=T' for panels (g)-(i) for the reader in the caption.

Line 220: I understand that wind on a massif scale isn't available, but can you provide more insight into prevailing wind patterns in this region in the context of your results?

Line 227-228: It is interesting that the slope angles differ slightly between the frequency categories defined here. As I previously mentioned, it may be worth examining the convexity as a function of frequency and deposit volume.

Line 243-244: As currently written, this sentence is a bit confusing. Do you mean that variations in snowpack characteristics due to changes in climate variables are more influential in determining deposit volumes than morphological variables? Please clarify.

Supplementary Figure 1: should "O" be "W" in x-axis labels?

---

## Referee Comment (RC2)

[referee-annotated manuscript omitted]

---

## Author Comment (AC1)

**Reviewer 1 comments and authors response**

**Paper title**: Brief communication: Weak control of snow avalanche deposit volumes by paths morphological characteristics

**Authors**: Hippolyte Kern, Nicolas Eckert, Vincent Jomelli, Delphine Grancher, Michael Deschatres, Gilles Arnaud-Fassetta

**Paper number**: tc-2021-103

We thank Reviewer 1 for his/her useful questions and comments on our manuscript. Please find below detailed feedback to individual comments and questions.

**Major comments:**

1.  Reviewer 1 highlights that *« to emphasize that the purpose of this study is to examine morphological variables exclusively the authors should explicitly state this as the objective in the Introduction »*

    We thanks R1 for this helpful suggestion. We will follow the recommandation and we will add the following sentence in the introduction to clarify the main objective of our study: « Here, the objective is to exclusively examine the relationship between avalanche path morphology and snow avalanche deposit volumes. »

2.  Reviewer 1 highlights that *« My biggest concern is the limited scope of inference that using just morphological variables to predict deposit volumes provides. However, I think the authors clearly state this limitation in the Abstract and Discussion and this work provides a solid quantitative measurement of the influence of morphological variables on deposit volume. »*

We agree with the fact that the general opinion supposes that morphological variables of the path have a limited impact on deposit volumes. We conducted this analysis to confirm and quantify this *a priori*. Consequently, the scope is indeed, only limited to the control of avalanche deposit volume by avalanche path morphology. In the revised paper we will address this concern by making the goal of our study even clearer as it was. Specifically we will modify several points in our discussion to further explain that the avalanche path geomorphology is indeed a rather minor, yet significant factor, for predicting depositional volumes and suggest potential explanations (e.g. climatic contribution) for the remaining variability of deposit volumes.

**Specific Comments**

- *Line 14: The last sentence is a bit confusing in the way it is currently written. What do you mean by "weakness"?*

We want to express that overall, the power of morphological variables to predict snow avalanche deposit volumes remains somewhat limited. « weakness » will be changed to « limited » in the revised manuscript.

- *Line 28: Include "morphological" before "factors" to emphasize the use of morphological variables exclusively.*

Changes will be made according to the suggestion.

- *Line 29-30: Should read "snow avalanche deposits". Remove "volumes".*

Changes will be made according to the suggestion.

- *Line 41: Geometric size or destructive size?*

This refers to the geometric size, modification will be done.

- *Line 42: How was the depth of debris deposits calculated, specifically? Width and length seem relatively straightforward to estimate, particularly based on images after the event, but can you elaborate on how observers estimate depth from the designated vantage point(s)?*

The EPA operators are very familiar with the studied paths, including their snowpack-free morphology and systematically use the same predefined observation point, so as to maximize the accuracy of the estimation, especially the depth of the deposit. However, of course their visual estimate has some uncertainties. Two sentences will be added in the revised manuscript to clarify this point: « The EPA operators are very familiar with the studied paths, including their snowpack-free morphology and systematically use the same predefined observation point, so as to maximize the accuracy of the estimation, especially the depth of the deposit. The depth of the deposit remains however difficult to estimate as for safety reasons this is not based on direct measurements on the deposit.»

- *Line 59: Do you mean "including snow avalanche records for which we did not calculate volumes"?*

Exactly, changes will be made in the revised manuscript.

- *Line 60: Avalanche occurrence rates?*

Yes, modification will be made.

- *Line 63-69: The reporting of snow depth in this paragraph is a bit confusing to me. Is the mean annual snowpack at the end of February only 90 cm at 2740m? Then, on average, the snowpack height increases another 80 cm from March through the end of May to reach 170 cm? Line 63 states that the snowpack depth at this elevation regularly exceeds 200 cm, though. Please clarify.*

We did not specify that we are using mean values for the period 2003-2017. For example, the spring snowpack is on average 170 cm but regularly exceeds 200 cm. We will modify the text to clarify

these issues : « The data from two weather stations handled by Météo-France and located at elevations of 1715 m a.s.l. and 2740 m a.s.l. in Bessans for the period 2003-2017, respectively (Fig. 1), was analyzed in order to determine climate conditions having locally prevailed over the study period. This showed that the depth of the local snowpack regularly exceeds 50 cm at 1715 m a.s.l. and 200 cm at 2740 m a.s.l.. The winter (Nov-Feb.) season is characterized by a cold mean air temperature (-4°C at 1715 m a.s.l., -5.5°C at 2740 m a.s.l.), with heavy precipitation that nearly only fall in the form of snow but the mean depth of the snowpack remains relatively thin (90 cm at 2740 m a.s.l.). By contrast, the spring season is characterized by higher mean air temperatures (3.5°C at 1715 m a.s.l., -2°C at 2740 m a.s.l.) and the occurrence of significant daily warm spells (daily mean air temperature up to 25°C at 1715 m a.s.l), which favors the occurrence of rain on snow events and wet snow avalanches. The mean daily fresh snowfall is half as much as during the winter season, but, the mean snowpack remains thick (170 cm). »

- *Line 79: What is the exact accuracy of the DEM? 1m?*

Yes, we will remove meter to specify 1 meter.

- *Line 82: It seems that the shape of the starting zone (or thalweg of the starting zone) represented by convexity and/or concavity might be a useful variable as well. Did you examine this?*

No, but we totally agree with the referee, this is a potential area of improvement that we are taking into consideration. We are currently developing a GIS tool to clearly define the starting zones and consequently improve our study. However, we decided to not include this point because this approach is still in progress. Moreover we conclude in our work that no matter which descriptors are used, the control of the deposits volume by path morphology remains weak and for us our investigations are sufficient to state this with full confidence. To clarify this point in the discussion, we will add these sentences: « Additional morphological descriptors, such as convexity or concavity of the starting zone, could slightly improve the predictive power of the models. However, we suspect that no matter which descriptors are used, the control of the deposits volume by path morphology remains weak. »

- *Line 116: Change "carryout out" to "completed".*

Change will be done

- *Line 130/Table 1: This table should probably be moved to the Results section.*

Indeed, we will move the table 1 to the Results section.

- *Line 141: Is this statistically significant? If not, consider using 'substantial' to avoid confusion.*

No this is no statistically significant, modification will be made.

- *Line 147: stronger? Perhaps 'more abundant' is a better word choice.*

Indeed, correction will be made

- *Line 150-151: I assume that each avalanche deposit volume is treated/measured individually as opposed to measuring the cumulative debris volume in paths with >1 avalanche over the course of the winter. Please clarify either here or in the Methods section how the individual avalanche height component in each volume calculation was separated from the cumulative height in paths where subsequent avalanche debris "stacked" on top of older debris.*

An estimation was performed by observed for each event to avoid the possibility of an estimation based on a superposition of several deposits. However, in rare cases, the estimations may be biased by a complex deposit superposition. A sentence will be added in the method section: « The depth of the deposit remains however difficult to estimate as for safety reasons this is not based on direct measurements on the deposit. This is especially problematic in case of pre-existing successive deposits, but observers try to take such effects into account as much as possible when providing their visual estimates »

- *Line 153: See my previous comment on Table 1 and since you reference the values in Table 1 here, Table 1 should be moved to the Results section.*

We moved table 1 to the Results section

- *Line 158-159: text reads "...positive correlation of frequency with min slope (r = -0.24 p < 0.05). This is slightly confusing because the r value is negative indicating a negative correlation. I read this negative r value as such: as frequency decreases, minimum slope increases. It seems that what you mean is that higher frequencies are correlated with lower minimum slope angle. Is this correct? Please clarify for the reader.  Also, perhaps I missed it, but I don't see frequency related to the morphological variables in any table in the main manuscript or supplement. Please clarify and consider including frequency correlation values in the table as well.*

Indeed, the negative r value indicates a negative correlation, modification has been made to clarify the sentence. We will add the frequency correlation to the morphological variables in the Supplementary table 1.

- *Line 161: change "slopes" to "slope"*

Done

- *Line 201/Figure 2: Please define 'Y=T' for panels (g)-(i) for the reader in the caption.*

*Y is corresponding to the predictor (combination of topographic variables) and T is corresponding to the target (observed deposit volume). This information will be added to the caption of figure 2.*

- *Line 220: I understand that wind on a massif scale isn't available, but can you provide more insight into prevailing wind patterns in this region in the context of your results?*

No, we do not have access to prevailing wind data. However several adjustments will be made in the discussion section to clarify how wind effects are considered in our study: « This correlation shows that winter deposit volumes may be influenced by prevailing climatic conditions.

Specifically, we suspect that the significant influence of orientation reveals wind impacts. Thus a prevailing wind from the west during the winter season may cause large accumulations of snow on the east oriented hillside, later favoring important deposit volumes. Such hypothesis remains however speculative without direct wind measurements at high elevations. »

- *Line 227-228: It is interesting that the slope angles differ slightly between the frequency categories defined here. As I previously mentioned, it may be worth examining the convexity as a function of frequency and deposit volume.*

As said before, we agree with you and will add a sentence in the discussion section: « Additional morphological descriptors, such as convexity or concavity of the starting zone, could slightly improve the predictive power of the models. However, we suspect that no matter which descriptors are used, the control of the deposits volume by path morphology remains weak. »

- *Line 243-244: As currently written, this sentence is a bit confusing. Do you mean that variations in snowpack characteristics due to changes in climate variables are more influential in determining deposit volumes than morphological variables? Please clarify.*

Indeed we speculate that variations in the snowpack due to changes in climate variables are more decisive than morphological variables to determine deposit volumes. However, we also want to mention the possibility that the roughness of the ground may have a direct effect on the snowpack characteristics, later influencing the avalanche characteristics. That is why we present this question as « an insightful perspective for further work ».

- *Supplementary Figure 1: should "O" be "W" in x-axis labels?*

Change will be done

---

## Author Comment (AC2)

**Referee Karl W. Birkeland comments and authors response**

**Paper title**: Brief communication: Weak control of snow avalanche deposit volumes by paths morphological characteristics

**Authors**: Hippolyte Kern, Nicolas Eckert, Vincent Jomelli, Delphine Grancher, Michael Deschatres, Gilles Arnaud-Fassetta

**Paper number**: tc-2021-103

We thank Reviewer Karl W. Birkeland for his useful questions and comments on our manuscript. Please find below detailed feedback to individual comments and questions.

**Major comments:**

1.  K.B highlights that: *« the title of the article (and in many other places) the authors talk about the "paths morphological characteristics". Since this is possessive, I believe they meant to write "paths' morphological characteristics". I think an even better way to write this would be "the morphological characteristics of the avalanche paths" or "avalanche path morphology". So, the title could be "Weak control of snow avalanche deposit volumes by avalanche path morphology". I think an even better title would simply be "The relationship between snow avalanche deposit volumes and avalanche path morphology", but the authors can decide on what they like the best. »*

    We thanks K.B. for this helpful suggestion. We will follow K.B recommendation and changes the title to « Weak control of snow avalanche deposit volumes by avalanche path morphology ». Also, we will change « path morphological characteristics » to « avalanche path morphology » everywhere in the revised paper.

2.  K.B. pointed out that: *« the authors do not specify if whether avalanche mitigation with explosives takes place in any of these avalanche paths. Are all the avalanches in the dataset natural releases? Or are they all explosive triggered? Or is there some mix? This is an important distinction that would definitely affect the results, and that needs to be clearly stated early in the manuscript. It would also be important to note if any avalanche paths have other defense structures, like catching dams, that might affect deposit volumes.*

    We thank K.B. for these important remarks. Indeed, few avalanches are preventively triggered to protect the road network. However, according to the EPA database only 53 of the 1491 avalanches we analyzed were triggered by explosives, and removing them from the analyzed sample does not affect our conclusions. Concerning the defense structures, a few are present in our study area but, again, not "enough" to affect our results. One of the reasons that led us to select this study area is that avalanche activity there is probably the most natural still existing in the French Alps. A small paragraph will be added in the data and methodology section to

explicit these aspects: « A small part of the avalanches are preventively triggered to protect the road network. According to the EPA database only 53 of the 1491 avalanches were triggered by explosives. Also, few defense structures are present in the studied paths but not enough to significantly affect our analysis. All in all, avalanche activity in the study area is among the most natural ones still existing in the French Alps. »

3. K.B. mentions that: *« the Discussion section needs additional work before this paper is publishable. I believe the authors should better explain their results and cite references where appropriate. For example, the first paragraph of the discussion just lists the results without any discussion at all. So, in the first paragraph they should explain why it makes sense that they found relationships between path mean elevation and mean deposit volumes, and path surface area and mean deposit elevation. It seems to me that a simple explanation is that higher elevations typically receive more snow, so might be more likely to produce larger volumes, and that larger surface areas provide more snow to avalanche, which would also produce larger volumes. This is just one example, but in the attached PDF I have tried to provide other possible explanations and I have also urged the authors to think more about their results and how they might be able to better discuss and explain them. »*

We thank K.B for this helpful suggestion. Several adjustments will be made according to K.B. suggestions to better explain our results and improve the discussion. For example, concerning the relationship between elevation and snow deposit volumes, we will follow K.B suggestion and will clarify this point. Indeed, as suggests by K.B: the higher the starting zone is, the bigger quantity of snow is available. Moreover, the higher the vertical drop is, the larger snow may be accumulated during the flow. A sentence will be modified in the discussion section of the revised manuscript: « In the three cases, mean elevation is retained as a relevant predictor, which underlines the relevance of snow availability in relation to elevation concerning the determination of deposit volumes. »

4. K.B mentions that: *« in the discussion there are some inconsistencies. Most of them are pointed out in the attached PDF, but I will highlight one here. On line 237 the authors state that "avalanche deposit volumes do not seem that much affected by avalanche path size", but on line 210 it says that one of the best simple relationships exists between avalanche deposit volumes and avalanche path surface area. Which of these two statements is correct? »*

Thanks to K.B suggestions, several inconstancies specified in the specific comments will be corrected in the revised manuscript. Concerning the inconstancy highlighted by K.B: on line 210, we are referring to the simple relationship results. On line 237 we are discussing the overall results, including the stepwise linear regression and neural network. Only simple relationships show that deposit volumes are correlated to the surface area, that's why we moderate our statements by saying that « snow avalanche deposit volumes do not seem that much to be affected by avalanche path size ».

5. Finally, K.B. suggests that: *« there is not a thorough discussion of the different complicating factors that may be affecting avalanche deposit volumes but which are not covered by this*

*study. I can think of one such factor: The presence or absence of a big area of wind fetch to the windward direction of the avalanche path. Having good fetch would allow for more wind-blown snow to be deposited in an avalanche path and would therefore increase avalanche deposit volumes. I would imagine the authors could think of many other complicating factors that affect avalanche deposit volumes, and that likely reduced the strength of the relationships between path morphology and deposit volumes. It would be good to list and discuss these. »*

Following K.B suggestions concerning the fact that winter deposits might show a weak correlation with east aspects due to wind loading from westerly winds, we will add in the revised version of the manuscript the following additional discussion: « This correlation shows that winter deposit volumes may be influenced by prevailing climatic conditions. Specifically, we suspect that the significant influence of orientation reveals wind impacts. Thus a prevailing wind from the west during the winter season may cause large accumulations of snow on the east oriented hillside, later favoring important deposit volumes. Such hypothesis remains however speculative without direct wind measurements at high elevations. »

**Specific Comments**

• *Line 16: I don't think you've made a strong case for the mechanical thresholds to be a primary driving force behind deposit volumes.*

We are not completely sure to understand what the referee means. However, to clarify what we mean: we interpret the weak relationship between mean path deposit volumes and morphological variables to be partly due the predominant control of by climate conditions (inducing variations in snowpack characteristics) and mechanical constraints. More precisely, we suspect that the deposit volume is connected to the mechanical thresholds involved in avalanche triggering processes, which is primary related to the snow mass and stratigraphy. To make it simple as soon as a critical value is reached the avalanche is released, as discussed in details in the many papers of the avalanche community investigating in detail the complex processes involved in avalanche triggering. This primary control by stress ration makes influence of other factors (terrain, climate) less directly critical .

• *Line 27: I am not aware of any experimental sites in Canada.*

Indeed, there is no experimental sites in Canada. We will change the formulation to precise that data was from field measurement : « Research conducted on experimental sites in Switzerland (Sovilla et al., 2015; Kölher et al., 2018) or from Canadian, Japan and European Alps field survey (Mc Clung and Gauer, 2018) showed weak links between avalanche deposit size, path slope and avalanche maximum frontal speed. »

• *Line 54: You should mention here if any avalanche control work is conducted on any of these avalanche paths? Are all the avalanches natural releases? Or are some explosive triggered?*

No, a small part of the avalanches are preventively triggered to protect the road. However, according to the EPA database only 53 of the 1491 avalanches were triggered by explosive. A

sentence will be added in the revised manuscript to explicit this point : « A small part of the avalanches are preventively triggered to protect the road, according to the EPA database only 53 of the 1491 avalanches were triggered by explosive. »

- *Line 66: I'm not sure I understand? In the previous sentence you state that the snowpack at 2740 "regularly exceeds 200 cm", but here you say that the snowpack at this elevation remains thin (90 cm). These two statments appear to contradict each other.*

We did not specify that we are using mean values for the period 2003-2017. For example, the spring snowpack is on average 170 cm but regularly exceeds 200 cm. We will modify the text to clarify these issues.

- *Line 79: I am not sure what this means? Do you mean a 1 m DEM? If not, how accurate of a DEM was used?*

We will remove meter to specify 1 meter.

- *Line 85: Would this be the primary orientation? What if the starting zone is a bowl with multiple aspects? How is the aspect determined for a path like that?*

The aspect is determined via a GIS tool, each cell aspect is computed and a mean value of all the cell values is obtained. Indeed, each cardinal direction is refrying to the global orientation of the path. We are not yet able to provide more specific orientations. « orientation » will be replaced by « primary orientation » as suggested by K.B

- *Line 139: Here are you referring to the mean deposit volume for the years 2003 and 2004? If so, add the word « mean ».*

Yes, we will add « mean »

- *Line 140: Again, is this the mean?*

Yes, we will add « mean »

- *Line 158: But aren't you showing a negative correlation with min slope and then a positive correlation with max slope?*

Indeed, the negative r value indicate a negative correlation, modification will be made to clarify the sentence.

- *Line 162: Which variables did you remove when doing the stepwise regressions? You said you would remove a variable if the Pearson p>0,8 between two variables, but you do not tell us which ones were removed.*

Because of a Pearson p>0.8 between max elevation and vertical drop, we removed them when doing the stepwise regressions. We will include the following sentence in the revised manuscript to

explicit this point: « Max elevation and vertical drop were removed as they were too strongly correlated. »

• *Line 164: Are min and mean elevation strongly correlated? Or not? Perhaps including a correlation matrix with all the correlations would be helpful?*

The correlation Matrix is present in the supplements (Supplementary Table 2). Min elevation and Mean elevation present a Pearson p of 0.26. These two variables are not strongly correlated. We will add the max elevation and vertical drop to the matrix correlation in the Supplementary table 2.

• *Line 165: Same as above. Are these three strongly correlated to each other? It seems like they might be?*

The three slope variables are not strongly correlated, the Pearson p values are below 0.5 (Supplementary Table 2)

• *Line 210: Why do you think this is the case? To me, these results make sense. We might expect to get larger avalanche deposit volumes in avalanche paths that are higher in elevation (so typically more snowfall) and have a larger surface area (more area for snow to accumulate before avalanching).*

These results are, indeed, intuitive; We share the same explanation: a large surface at high altitude favors important snow accumulation and large deposits. A sentence will be modified to clarify this explanation: « The best simple relationships were observed with path mean elevation (r = 0.51) and surface area (r = 0.48): a large surface at high altitude favors important snow accumulation and large deposits. »

• *Line 212: Why do you think this is the case? What can you think of that might help explain this?*

We think that the low value for spring simple correlation between deposit volumes and avalanche path morphology may be due to an important control of climate conditions. However, we thought that it was better to present this aspect of the discussion after presenting stepwise and neural networks results and discussion. However, as suggested by K.B, we will add a sentence to discuss this result: « This may be due to climate conditions that may strongly control spring deposit volumes (e.g. wet snow avalanches are released as soon as cohesion drops within the snowpack due to the apparition sufficient liquid water, and rather independently of the snow mass. »

• *Line 213: Here I am missing some discussion. You have presented results in this paragraph, but you have not discussed those results. Why do you think you found the relationships you found? Can you guess at some possible explanations? When I read this I think it makes sense. I wold think that winter deposits might show a weak correlation with east aspects due to wind loading from westerly winds. Do you think this is the case? Or, do you have some other possible explanations?*

Indeed, we agree. A prevailing wind from the west accumulating snow on the east oriented hillside may explain this correlation. Few sentences will be included to explicit this point: « This correlation

shows that winter deposit volumes may be influenced by prevailing climatic conditions. Specifically, we suspect that the significant influence of orientation reveals wind impacts. Thus a prevailing wind from the west during the winter season may cause large accumulations of snow on the east oriented hillside, later favoring important deposit volumes. Such hypothesis remains however speculative without direct wind measurements at high elevations. »

• *Line 217: Why do you think this is the case? Perhaps due to snow availability at different elevations?*

We agree with your explanation, the higher the starting zone is, the bigger quantity of snow is available. Moreover, the higher the vertical drop is, the larger snow may be accumulated during the flow. A sentence will be modified in the discussion section of the revised manuscript: « In the three cases, mean elevation is retained as a relevant predictor, which underlines the relevance of snow availability in relation to elevation concerning the determination of deposit volumes. »

• *Line 218: This seems unusual. Can you explain it? The reason it seems unusual is that I don't know why East and West aspects would both be positively correlated with avalanche deposit volumes.*

A logical explanation for this result would be that the prevailing winds are either from the west or from the east. We know for example that important storms coming from the east can cause important snow accumulation in this study area. However, without reliable data concerning the wind, it is not possible to validate this hypothesis.

• *Line 219: Yes, this is true, but explain how solar radiation and wind affect deposit volumes.*

We will change this to clarify the effect of solar radiation and wind on the snowpack with the following sentence: « This indicates how important the solar radiation and/or the path positioning in respect to the prevailing wind direction may be to generate the snowpack and then produce instabilities, later influencing volume deposits. »

• *Line 221: This is true, but wouldn't you expect larger volumes to be coming out of paths facing east and southeast if your main upper air winds and storms come from the west and northwest? Also, even though you cannot precisely define the wind direction and speed, you can characterize the overall upper air winds which help to control those local wind patterns.*

Indeed this could be an intuitive speculation. However the mean deposit volumes are similar between east and west oriented paths. We also think the prevailing wind direction may not be the only explanation concerning the wind influence on deposit volumes. For example, a winter storm coming from the east may have more influence on the deposit volumes than a global west prevailing wind. Because our data are from annual or seasonal average, we prefer not to make any over-interpretation based on shorter time periods than those considered here.

• *Line 223: Why do you think this is the case? Too many other variables? Or some other cause?*

We will add two sentences to clearly explain that, no matter the geomorphological variables, the control of the deposits volume by path morphology remains weak: « Additional morphological descriptors, such as convexity or concavity of the starting zone, could slightly improve the predictive power of the models. However, we suspect that no matter which descriptors are used, the control of the deposits volume by path morphology remains weak. »

As said in last part of the discussion, we speculate that the weak relationship between volume and morphological variables may be due to a stronger influence of climate conditions than morphological variables.

- *Line 224-225: Do you have any evidence that the relationships are non-linear?*

We speculate that the relationships between deposit volumes and path morphology might be non-linear because of the non-linear process involved in avalanche triggering. This speculation seems to be in line with the neural networks results that overpass linear models in terms of predictive power.

- *Line 230: This is counterintuitive. One would think that a path that released more often would have a smaller volume for each release. But, here that seems not to be the case? Do you have any explanations why? Could it be that some avalanche paths are simply better situated due to local topography to collect more snow? So, those paths both run more frequently and produce a greater volume of avalanche debris?*

Indeed this is counterintuitive, one explanation may be related to the threshold selected (about two avalanches per year) to distinguish very active paths. This threshold would be too low to remove all the snow in the catchment between two events. To exclude the paths that present regular purging phenomena (and may indeed show a negative correlation between volumes and frequency), which could reduce the mean deposit volume, the frequency threshold should be higher. Our data are too limited to consider the paths that present regular purging phenomena.

- *Line 231: What did Sovilla et al. (2010) find? Did they also find that avalanche paths that run more often also produce bigger volumes of snow?*

No, Sovilla et al. (2010) highlighted a negative correlation between deposit depth and slope angle in the deposit area, however, they also observe a complex relationship with the frequency which itself is determined by the slope of the path. A sentence will be modified to explicit this point: «These somewhat counterintuitive results are in line with those of Sovilla et al. (2010) that highlighted a negative correlation between slope angle and deposit depths, partly affected by the avalanche activity. »

- *Line 237: Just above in line 210 you state that avalanche path surface area is strongly correlated to deposit volumes. But here you say they are not affected by avalanche path size. Which of these two is actually correct?*

On line 210, we are referring to the simple relationship results. On line 237 we are discussing the overall results, including the stepwise linear regression and neural network. Only simple

relationships show that deposit volumes are correlated to the surface area, that's why we moderate our statements by saying that « snow avalanche deposit volumes do not seem that much to be affected by avalanche path size ».

- *Line 238: This seems logical to me, but you should explain why avalanches in the spring might produce bigger or smaller deposit volumes than in the winter.*

A sentence will be included to explicit the differences between winter and spring mean volumes: « Differences in the snowpack characteristics may also explain why the winter deposit mean volumes present more important values. Indeed the winter snowpack is less stable and prone to large avalanche triggering. In other words, snow storms are frequent in winter and favor major instabilities and large snow avalanches. »

- *Line 238-240: I don't understand how these explain the differences in snow avalanche deposit volumes. For this region I would assume that most zones with a similar aspect and elevation would have a similar snow depth and stratigraphy. Thus, aspect and elevation would be indirect proxies of snow depth and stratigraphy. Of course, there is a lot of noise in these relationships, However, in the end I wouldn't think that snow mechanical behavior would have a dramatic effect on deposit volumes in this area since you are working with a relatively small area. Certainly this would be a factor if you were comparing this area to an area with a different snow climate. Can you explain further how you think mechanical properties of snow could be affecting deposit volumes ?*

Indeed, the snow depth and stratigraphy is similar within our study area within release zones with the same aspect and elevation. However, the snow mechanical behavior is partly defined by other variables, such as slope, curvature, etc. The morphology of studied paths shows strong variations in these variables , and we are using a particularly large dataset of 1450 avalanches. So, it is logical that even for close release elevations and aspects there is a strong variability in deposit volumes reflecting the influence of mechanical variables involved in the avalanche triggering.

- *Line 245: While it is true that climate conditions will affect avalanche volumes, I don't see how this can be applied to your work. In your study you use a group of 77 avalanche paths from the same area. So, changes in climate should - mostly - affect all the paths similarly. Thus, one would still expect to see that the avalanche path morphology would have a more significant impact on the deposit volumes. Of course, there are many complicating factors and this is likely why the strength of your relationships is relatively weak.*

We thank K.B for these important remarks. We agree with you, changes in the climate should vastly affect the paths similarly. Indeed, many factors may explain our weak strength of our relationships. That's why we propose to extend our approach by investigating both the meteorological and morphological variables and extending our study area. We will modify the outlook section of the discussion to emphasize the importance of extending the study area: « Our approach should therefore now be extended to simultaneously take into account the control of deposit volumes by

morphological and meteorological variables on a wider study area, and how these controls evolve as climate change goes on. »

However, we are speculating that even minor changes in the climate, caused by differences in elevation and aspect, can cause major instability in the snowpack, later influencing the deposit volumes.

- *Line 249: I was wondering why you did not attempt to look at how different meteorological factors affected avalanche deposit volumes. Did you feel that was beyond the scope of this paper?Even something simple like looking at total snowfall recorded at one of the weather stations and whether or not yearly changes were - or were not - correlated with yearly changes in deposit volumes would be interesting. And this would be a first step in figuring out how climate change might affect avalanche deposit volumes.*

We totally agree with you, and we are currently working on analysis based on meteorological and geomorphological variables. However, as said before this is a complex point because of many interconnected factors. That is why we thought it was beyond the scope of this paper.

Minor corrections and typographical errors identified by K.B has been corrected.